# One Year of COVID-19: Lessons Learned in a Hand Trauma Center

**DOI:** 10.3390/jcm11082163

**Published:** 2022-04-13

**Authors:** Marcello Covino, Camillo Fulchignoni, Silvia Pietramala, Marco Barbaliscia, Francesco Franceschi, Giulio Maccauro, Gianfranco Merendi, Lorenzo Rocchi

**Affiliations:** 1Department of Emergency Medicine, Fondazione Policlinico Universitario A. Gemelli, IRCCS—Catholic University of the Sacred Hearth, 00168 Rome, Italy; macovino@gmail.com (M.C.); francesco.franceschi@policlinicogemelli.it (F.F.); 2Orthopedics & Hand Surgery Unit, Department of Orthopedics, Fondazione Policlinico Universitario A. Gemelli, IRCCS—Catholic University of the Sacred Hearth, 00168 Rome, Italy; sy.pietramala@gmail.com (S.P.); marcobarbaliscia@outlook.it (M.B.); gianfrancomerendi@icloud.com (G.M.); lorenzo.rocchi@policlinicogemelli.it (L.R.); 3Orthopedics & Traumatology Unit, Department of Orthopedics, Fondazione Policlinico Universitario A. Gemelli, IRCCS—Catholic University of the Sacred Hearth, 00168 Rome, Italy; giulio.maccauro@policlinicogemelli.it

**Keywords:** hand surgery, COVID, lockdown, emergency department, patient admission

## Abstract

2020 will be remembered worldwide as the year of COVID-19 outbreak. The onset of this pandemic abruptly changed everybody’s life and, in a particular manner, doctors’ lives. Our hand surgery department became rapidly one of the first COVID-19-specialized wards in Italy, impacting considerably the authors’ routines and activities. In this paper, the authors focus on how the demographics of patients with hand trauma changed and how they had to modify their activity. The authors retrospectively took into consideration all patients reaching their emergency department (ED) with hand trauma between 9 March 2020 (the day of the beginning of the first lockdown in Italy) and 8 March 2021 and compared them to those who reached the ED in the three previous years. Authors have analyzed the number of patients, their gender and age, the severity of their trauma, where the trauma occurred, the type of lesion, the percentage of patients who underwent surgery, and the percentage of patients who had an emergency admission. In the last year, the number of patients reaching the ED for a hand trauma has been reduced by two thirds (975 patients during the past year), the mean age of those patients has slightly increased, the severity of cases has increased, places of trauma and type of lesions have changed, and, lastly, the percentage of patients needing surgery who were admitted immediately has increased. This paper shows how the type of patients reaching the ED changed and discusses how surgeons evolved and modified their habits in treating those patients during the first lockdown and the year that followed.

## 1. Introduction

In 2015, Giustini et al. [1] estimated that, in Italy, upper extremity injuries represent approximately 20% of all emergency department (ED) admissions for injury. Among those, almost 60% (equivalent to more than 880,000 patients per year) access the ED for a hand problem, making those injuries some of those more frequently encountered and treated in Italian emergency departments.

At the end of 2019, a new coronavirus was described in China in patients with pneumonia [2]. The rapid worldwide spread due to the high contagiousness and the elevated rate of hospitalization and death due to what has rapidly come to be called by the World Health Organization the “COVID-19 pandemic” led to general lockdowns all over the world. In Italy, the first general lockdown lasted from 9 March 2020 to 4 May 2020, followed by different periods of various restrictions, depending on local decisions, along with the different COVID waves. The COVID-19 pandemic and the relative restrictions have not only affected everybody’s daily life, industry, and economics, but also healthcare providers worldwide [3,4,5].

With the restrictions in place, the authors immediately noted that the number of patients reaching their ED for hand trauma problems decreased, as the type and severity of lesion changed. Therefore, they decided to retrospectively analyze all patients joining their ED, studying the type and gravity of lesions and their treatment not only during the first lockdown, but during the whole year that followed the beginning of the first quarantine. Those data are compared to those of the three previous years. 

To the authors’ knowledge, up to the present, there is no other article in the literature studying the modification in typology of patients with hand trauma during the whole year after the beginning of the first COVID-19 lockdown.

## 2. Materials and Methods

Authors searched for the keywords “hand” and “finger” on the dismissal diagnosis available on their ED software for all patients admitted between 9 March 2020 and 8 March 2021 and for the three previous years: 9 March 2019–8 March 2020, 9 March 2018–8 March 2019, and 9 March 2017–8 March 2018. The clinical records of the extracted patients were manually reviewed by the authors.

Collected information for each patient included: Age;Sex;Triage severity code: increasing from white and green (ambulatorial) to yellow (urgency) and red (emergency);Days of prognosis;Cause of the accident: domestic, work, school, traffic, sport, other (including self-harm lesions, bites, assaults, non-specified);Type of lesion: amputation, fracture, tendinous, other (including nerves, dislocation, bruise, cutaneous, infection, non-specified);If the patient underwent surgery for the hand lesion that brought them to the ED;If the patient was hospitalized directly from the ED.

Continuous variables (age and prognosis) are presented as means, ranges, and standard deviations (SD). The significance of differences between means was assessed using the *t*-statistic calculated as part of the two-tailed *t*-Student test with a confidence level of 95%, significance level α = 0.05. The significance of the evolution of bounded categories (color code, cause of accident, type of lesion, surgery, and hospitalization) was assessed by comparing their distribution (9 March 2020–8 March 2021 versus a mean of the 3 previous years) using a two-sided chi-square test with a confidence level of 95%, significance level α = 0.05.

## 3. Results

### 3.1. Accesses to ED

During the year following the first day of lockdown due to the COVID-19 pandemic, from 9 March 2020 to 8 March 2021, 974 patients accessed the authors’ ED for a hand problem. This is just above half when compared to the same period of the three previous years: 1848 (2019–2020), 1872 (2018–2019), 1785 (2017–2018). The difference is even greater when focusing on the first lockdown period (from 9 March 2020 to 4 May 2020) when 90 patients reached the ED for a hand problem versus 335 in 2019, 305 in 2018, and 339 in 2019.

### 3.2. Age and Sex

The age and sex of patients are reported in Table 1. The distribution of sex of patients reaching the authors’ ED with a hand problem did not change during the COVID-19 pandemic, with males representing two thirds of all patients. Regarding mean age, it was higher during the period going from 9 March 2020 to 8 March 2021 when compared to the mean age of patients during the three previous years in a statistically significant manner (*p* < 0.01).

### 3.3. Severity of Lesions

The severity of hand lesions has been assessed by the mean of the severity color code given when the patient reached the ED, and by the prognosis days given at dismissal. It appears that patients who came to the ED during the COVID-19 pandemic had more serious injuries when compared with the previous years. Mean days of prognosis increased from 8.2 (range 0–90, SD 11.3) before COVID-19 to 10.8 (range 0–60, SD 12.8) during the pandemic; this was statistically significant (*p* < 0.01). This worsening of severity in hand lesions reaching our ED is also expressed significantly (*p* < 0.01) by the color code given at the ED (Figure 1).

### 3.4. Causes and Types of Lesions

Causes of hand injuries are reported in Table 2 and Figure 2; types of lesions are reported in Table 3. 

The rate of school- and sports-related incidents have decreased during the pandemic, which can easily be linked with the limitation of sports practice and with the predominance of homeschooling during this period, whereas the domestic incidents rate has increased. Those distribution modifications were statistically significant (*p* < 0.01). 

Regarding the type of lesions: amputations [6], fractures [7], and tendinous lesion [8] rates slightly increased during the pandemic; but in a non-statistically significant manner (*p* = 0.12).

### 3.5. Treatment and Hospitalization

Among the patients who reached the ED with a hand problem during the 9 March 2020–8 March 2021 period, 43.8% underwent surgery, which is a statistically higher rate (*p* = 0.01) than those needing surgery (32.0%) in the three years before the pandemic. This confirms that patients who came to the ED since the beginning of the lockdown had more severe injuries.

Regarding hospitalization rates: 17.6% of patients reaching the authors’ ED for a hand problem after 9 March 2020 were directly hospitalized from there; they represent 40.0% of the surgically treated patients. Before that date, 8.5% of all the patients were directly hospitalized from the ED and they represented 26.7% of the surgically treated patients. This statistically significant increase (in both cases *p* < 0.01) is more likely due to organizational issues.

## 4. Discussion

The outbreak of the COVID-19 pandemic at the beginning of 2020 and the subsequent lockdown and limitations have dramatically changed not only everybody’s life, but all professional sectors. Healthcare workers were not immune to this tragic event, nor were hand surgery centers. All around the world, rapid changes occurred in the organization of wards and activities to adapt to new necessities [9,10,11,12,13,14,15]. In best-case scenarios, hand surgeons were required to adapt to a reduction of hand surgery activity, whereas, in other cases, some colleagues had to change jobs to help in COVID wards.

Many hand surgery teams have already published their experiences, but all of them focused only on the first lockdown period. Despite this, many found results similar to the authors’. Poggetti et al. [16] analyzed only patients who underwent surgery; they had a diminution in the absolute number of patients treated (168 in 2019 versus 120 in 2020) with patients being on average older, a diminution of traffic- and work-related accidents, and an increase in work and domestic injuries. The only statistical difference in the type of lesion was an increase in patients with a fracture of the proximal and middle phalanges. Atia et al. [17], discussing all cases reaching their ED, presented a diminution in patients (793 in 2019 versus 463 in 2020) who were also on average older, an increase in intervention rate (41% in 2020 versus 30% in 2019), a diminution in the mean time to surgery, and no difference in the type of cases. Pichard et al. [18] had 275 patients reaching their hand center in the lockdown period of 2020 (versus 784 on the same days of the year before) and patients were older in comparison to 2019′s patients. There was a decrease in work, leisure, and traffic incidents with an increase of domestic lesions. They also described an increase in soft tissue lesions with a decrease in fractures and a higher surgical rate (51.2% versus 36.9%). Fortaine [19] and Pidgeon [20] both described a diminution of 52% of patients reaching their ED, with Fortaine’s group also giving details of how sport and work incidents diminished, whereas domestic injuries increased, and how there was a global increase in severity of lesions. Regas et al. [21] also mentioned how hand and wrist lesions were less frequent but more severe. Finally, Ho et al. [13] are the only group with an increase in the number of cases reaching their ED (332 versus 310), this is likely due to the fact that patients from another hospital were automatically transferred to theirs during the lockdown period. Overall, except for this last case, every group had a diminution in the number of patients, although none had a 75% decrease similar to that the authors of this article had when referring to the two months of lockdown. As in most of the papers, the authors had older patients, with a decrease in school and sports injuries and an increase in domestic lesions, whereas traffic and work injuries did not change. Most probably, those results can be explained by the fact that the authors analyzed numbers for the full year following the initial lockdown when homeschooling and sports limitations were underway, but people were allowed to go back to work and use their cars. Also, as in the two papers mentioning the severity of ED injuries [19,21], after the outbreak of COVID, injuries were more severe in the authors’ EDs. Moreover, in the same period, the authors had an increase in intervention rates such as reported by Atia [17] and Pichard [18]. 

The increase in the authors’ overnight hospitalization rate is not only explainable by the increase in severity of cases, but also by the necessity to change habits as it became more difficult to hospitalize patients a second time as the authors often used to do. Other changes were brought, including the use of absorbable skin sutures and less frequent check-ups with patients when possible. Additionally, Atia et al. [17] described checking up on patients less frequently, whereas most of authors used telemedicine to follow up with their patients [12,14,22,23,24]. This is a method that proved its accuracy [25] and found satisfaction among patients [10,22,26], whereas surgeons had frequent issues [10,22]. The authors did not use telemedicine, and Toia et al. [11] as well as Leti Acciaro et al. [27] confirmed it might be an issue in Italy. To reduce contamination risks [28], some surgeons increased non-surgical treatment [9,23], and to resolve issues linked to a diminution of anesthetists’ availability, many hand surgeons [9,12,14,17,24,29] stated to use WALANT surgery [30].

Another issue not directly proven by this study but directly linked with the diminution of patients coming to the emergency department and the organizational problems with hospitalizing patients [27,31] is the risk to increase late diagnosis and late treatment [32] in patients, which can be proven by the augmentation of scaphoid non-union [33]. This latter point could also, in some cases, be related to patients being afraid to access the emergency department because of the risk of contamination [34]. In the same way, another issue to resolve in the near future, is the risk of patients receiving delayed rehabilitation [11].

Finally, being from a teaching hospital, it is difficult not to speak of the impact that the COVID-19 pandemic might have had on resident physicians. As in many other hospitals [10,11,14,19], the authors had all their elective surgery canceled during the first months after the beginning of the lockdown, and still highly diminished during the year that followed, creating a significant lack in young hand surgeons’ formative experience. In some cases [9], residents were allowed limited access to the hospital. Lastly, at the authors’ university, as at others [9,14], lessons switched from in situ to webinars, leading to a decreased possibility of interaction between professors and residents.

## 5. Conclusions

This study has some limitations, including its retrospective design and the fact that collection of data was performed only from ED clinical charts. Nevertheless, the authors believe that this study gives good input and ideas for the future; although the COVID-19 outbreak led to a change in etiologies and frequency of hand trauma, the number of patients reaching ED has remained high, confirming the importance of maintaining a fully functioning hand department during an eventual future pandemic. Authors also believe that the government should increase the prevention of domestic injuries to avoid further increases of lesions in this area if people will be newly obliged to stay home. Finally, as also mentioned by Toia et al. [11], the Italian government should invest in telemedicine to allow Italian surgeons to treat their patients equally as in other countries if further pandemics occur.

## Figures and Tables

**Figure 1 jcm-11-02163-f001:**
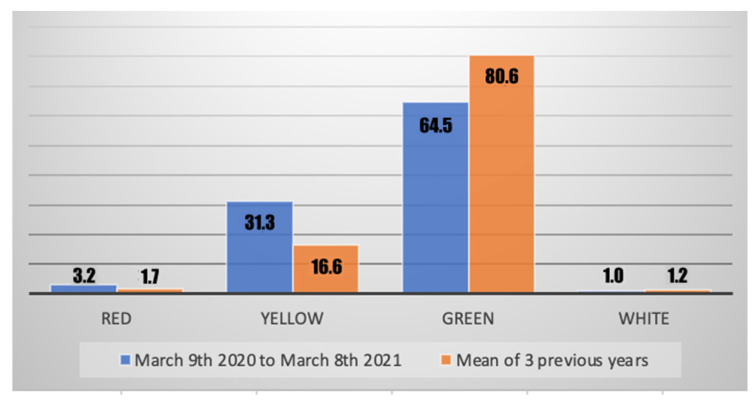
Comparison of severity color code (in percent) of hand injuries reaching our ED before and during the COVID-19 pandemic.

**Figure 2 jcm-11-02163-f002:**

Causes/places of hand injuries of people reaching the authors’ ED before and during the COVID-19 pandemic.

**Table 1 jcm-11-02163-t001:** Demographics.

		9 March 2020 to 8 March 2021	9 March 2019 to 8 March 2020	9 March 2018 to 8 March 2019	9 March 2017 to 8 March 2018	Mean of Previous 3 Years
Total number of patients		974	1848	1872	1785	1835
Mean age of patients *		39.6	35.1	35.3	34.8	35.1
(range)		(0–96)	(0–92)	(0–96)	(0–93)	(0–96)
(Standard deviation)		(21.0)	(21.8)	(22.0)	(22.0)	(21.9)
Sex	Male	653 (67.0%)	1212 (65.6%)	1208 (64.5%)	1152 (64.5%)	1190.7 (64.9%)
	Female	321 (33.0%)	636 (34.4%)	664 (35.5%)	633 (35.5%)	644.3 (35.1%)

* In years.

**Table 2 jcm-11-02163-t002:** Cause/location of accidents.

	9 March 2020 to 8 March 2021	9 March 2019 to 8 March 2020	9 March 2018 to 8 March 2019	9 March 2017 to 8 March 2018	Mean of Previous 3 Years
Domestic	310 (31.8%)	404 (21.9%)	418 (22.3%)	395 (22.1%)	405.7 (22.1%)
Work	98 (10.1%)	170 (9.2%)	145 (7.7%)	159 (8.9%)	158 (8.6%)
School	22 (2.3%)	92 (5.0%)	90 (4.8%)	83 (4.7%)	88.3 (4.8%)
Traffic	89 (9.1%)	186 (10.1%)	165 (8.8%)	171 (9.6%)	174 (9.2%)
Sport	36 (3.7%)	172 (9.3%)	181 (9.7%)	154 (8.6%)	169 (9.2%)
Other	419 (43.0%)	824 (44.5%)	873 (46.7%)	823 (46.1%)	840 (45.8%)

**Table 3 jcm-11-02163-t003:** Type of lesions.

	9 March 2020 to 8 March 2021	9 March 2019 to 8 March 2020	9 March 2018 to 8 March 2019	9 March 2017 to 8 March 2018	Mean of Previous Years
Amputations	62 (6.4%)	98 (5.3%)	96 (5.1%)	83 (4.7%)	92.3 (5.0%)
Fractures	256 (25.2%)	377 (20.4%)	385 (20.6%)	334 (18.7%)	365.3 (19.9%)
Tendinous lesions	59 (6.1%)	50 (2.7%)	55 (2.9%)	70 (3.9%)	58.3 (3.1%)
Other	607 (62.3%)	1323 (71.6%)	1336 (71.4%)	1298 (72.7%)	1319 (71.9%)

## Data Availability

Data is available on demand; contact the corresponding author.

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
