# Peer review of "One Year of COVID-19: Lessons Learned in a Hand Trauma Center"

_jcm, 2022, doi:10.3390/jcm11082163_

Round 1

Reviewer 1 Report

The authors describe the patients presenting to the emergency room during the COVID lockdown. Since health systems were so overwhelmed, limiting our response, this is important information in planning for future pandemics/similar situations.

It may be beneficial to describe some elements of Italy’s health system and how they:

1) failed

2) could be improved in preparation for another wave or different situation.

This may help other health systems prepare better beyond the Italian health system

Author Response

POINT 1: Moderate English changes required

Response 1: English has been edited.

POINT 2:

It may be beneficial to describe some elements of Italy’s health system and how they:

1) failed

2) could be improved in preparation for another wave or different situation.

This may help other health systems prepare better beyond the Italian health system

Response 2: The authors don't feel in the capacity to judge how and where the Italian heath system may have failed facing Covid-19 without the necessary hindsight. Regarding the second point, the authors already gave in the conclusion of this paper their suggestions on how they believe the system could be improved in preparation of another pandemic. 

Reviewer 2 Report

Dear,

According to the review of: One year of COVID-19: how the type of patients……”

- The Title may be :  One year of COVID-19: how the typologies of patients with hand trauma was impacted

- Lines19-20: “We focus in this paper….our activity.” May be improved in: “This paper focused on the demographic changing of the patients with hand trauma and how clinical an surgical activities were modified.”

We feel the sentence  “The  authors” may be better than “We” in all the paper.

- Lines 56-58: The sentence is wrong. There are other papers published and on line first ti cite: Change the sentence in “Few artcile in literature described the modification of patient typology with hand trauma during the whole year after the beginning of the first COVID-19 lockdown [5,6].” Add 5 and 6 in the citations as following

Leti Acciaro A, Montanari S, Venturelli M, et al. New management and trauma incidence in hand surgery during the phase 1 of COVID-19 pandemic in a referral hand surgery and microsurgery center into the outbreak in North Italy. Minerva Orthopedics 2021;72(5):437-9. doi:10.23736/S2784-8469.20.04075-8

6  Leti Acciaro A, Montanari S, Venturelli M, et al. Retrospective study in clinical governance and financing system impacts of the COVID-19 pandemic in the hand surgery and microsurgery HUB center. Oublished on line first: 2 Feb 2021. doi: 10.1007/s12306-021-00700-3

The actual citation number 5 will be 7 as well as the others will be upgrade ...

- Line 193: add: “and Toia et al. [12] as well as Leti Acciaro et al. [5] confirmed it ..”

- Line 198: add the new citations 5 and 6 in the text  “and organizational problems to hospitalizing patients [5,6] is the risk...”

Author Response

Point 1: Moderate English changes required

Response 1: English has been edited

Point 2: The Title may be :  One year of COVID-19: how the typologies of patients with hand trauma was impacted

Response 2: title has been modified

Point 3: Lines19-20: “We focus in this paper….our activity.” May be improved in: “This paper focused on the demographic changing of the patients with hand trauma and how clinical an surgical activities were modified.”

We feel the sentence  “The  authors” may be better than “We” in all the paper.

Response 3: "we" was changed with "the authors" through out the entire manuscript

Point 4: Lines 56-58: The sentence is wrong. There are other papers published and on line first ti cite: Change the sentence in “Few artcile in literature described the modification of patient typology with hand trauma during the whole year after the beginning of the first COVID-19 lockdown [5,6].”

Add 5 and 6 in the citations as following

5  Leti Acciaro A, Montanari S, Venturelli M, et al. New management and trauma incidence in hand surgery during the phase 1 of COVID-19 pandemic in a referral hand surgery and microsurgery center into the outbreak in North Italy. Minerva Orthopedics 2021;72(5):437-9. doi:10.23736/S2784-8469.20.04075-8

6  Leti Acciaro A, Montanari S, Venturelli M, et al. Retrospective study in clinical governance and financing system impacts of the COVID-19 pandemic in the hand surgery and microsurgery HUB center. Oublished on line first: 2 Feb 2021. doi: 10.1007/s12306-021-00700-3

Response 4: the authors did not change the sentence  as they still believe there are no other papers discussing  modification in typology of patients with a hand trauma during the whole year after the beginning of the first COVID-19 lockdown. The two articles mentioned above discuss only a 2 months period. Nevertheless, both articles have been included as references as suggested below:

- Line 193: add: “and Toia et al. [12] as well as Leti Acciaro et al. [5] confirmed it ..”

- Line 198: add the new citations 5 and 6 in the text  “and organizational problems to hospitalizing patients [5,6] is the risk...”

Reviewer 3 Report

Thankyou for giving me the possibility to review the paper ginal "One year of Covid-19: how the type of patients getting to the Emergency Department for a hand trauma was impacted". 

The present paper, however, has several concerns and just shows the results already reported by a huge number of papers. 

The paper is not innovative neither originali.

Author Response

Point 1: Moderate English changes required

Response 1: English has been edited

Point 2 : The present paper, however, has several concerns and just shows the results already reported by a huge number of papers. The paper is not innovative neither originali.

Response 2: Beyond offering a review of the literature on the subject, this paper is the first to analyze the modification in typology of patients with a hand trauma during the whole year after the beginning of the first COVID-19 lockdown.